# Inhibition of Digestive Enzymes and Antioxidant Activity of Extracts from Fruits of *Cornus alba*, *Cornus sanguinea* subsp. *hungarica* and *Cornus florida*–A Comparative Study

**DOI:** 10.3390/plants9010122

**Published:** 2020-01-18

**Authors:** Joanna Truba, Iwona Stanisławska, Marta Walasek, Wioleta Wieczorkowska, Konrad Woliński, Tina Buchholz, Matthias F. Melzig, Monika E. Czerwińska

**Affiliations:** 1Student Scientific Association, Department of Pharmacognosy and Molecular Basis of Phytotherapy, Medical University of Warsaw, 1 Banacha Street, 02-097 Warsaw, Poland; asia.truba@gmail.com (J.T.); marwal9@wp.pl (M.W.); wiolasekutowicz@gmail.com (W.W.); 2Department of Bromatology, Medical University of Warsaw, 1 Banacha Street, 02-097 Warsaw, Poland; istanislawska@wum.edu.pl; 3Polish Academy of Sciences Botanical Garden, Center for Biological Diversity Conservation in Powsin, 2 Prawdziwka Street, 02-973 Warsaw, Poland; k.wolinski@obpan.pl; 4Institute of Pharmacy, Freie Universitaet Berlin, 2 + 4 Koenigin-Luise street, D-14195 Berlin, Germany; tina.buchhol@gmail.com (T.B.); matthias.melzig@fu-berlin.de (M.F.M.); 5Department of Pharmacognosy and Molecular Basis of Phytotherapy, Medical University of Warsaw, 1 Banacha Street, 02-097 Warsaw, Poland

**Keywords:** Siberian dogwood, polyphenols, digestive enzymes, diabetes, obesity

## Abstract

The fruits of some *Cornus* species (dogwoods) are used in traditional medicine and considered potential anti-diabetic and hypolipemic agents. The aim of the study was to determine the ability of extracts from *Cornus alba* (CA), *Cornus florida* (CF), and *Cornus sanguinea* (CS) to inhibit digestive enzymes namely *α*-amylase, pancreatic lipase, and *α*-glucosidase, as well as isolation of compounds from plant material with the strongest effect. In addition, the phytochemical profile and antioxidant activity of extracts from three dogwoods were compared with HPLC-DAD-MS/MS and DPPH scavenging assay, respectively. Among the aqueous-ethanolic extracts, the activity of *α*-amylase was the most strongly inhibited by the fruit extract of CA (IC_50_ = 115.20 ± 14.31 μg/mL) and the activity of *α*-glucosidase by the fruit of CF (IC_50_ = 38.87 ± 2.65 μg/mL). Some constituents of CA fruit extract, such as coumaroylquinic acid, kaempferol, and hydroxytyrosol derivatives, were isolated. Among the three species of dogwood studied, the greatest biological potential was demonstrated by CA extracts, which are sources of phenolic acids and flavonoid compounds. In contrast, iridoid compounds or flavonoid glycosides found in fruits of CF or CS extracts do not play a significant role in inhibiting digestive enzymes but exert antioxidant activity.

## 1. Introduction

The genus *Cornus* (dogwood) has been first described by Linnaeus and it was divided into five morphologically diverse species, which are typically recognized as major representatives of subgroups within this genus nowadays [1]. The dogwoods are mostly trees or shrubs, widely distributed in the northern regions of Europe, Asia, and North America. Species of *Cornus* L. are characterized by specific types of inflorescence and fruits. The phytochemical composition, taxonomy, and phylogenetic relationships within the genus *Cornus* L. sensu lato seem to be highly controversial and have been discussed for a long time [2]. Based on the data of chloroplast genome and morphological characters five major lineages, such as (1) the alternate-leaved, blue-fruited dogwoods, (2) the opposite-leaved, blue- or white-fruited dogwoods, (3) the cornelian cherries, (4) the dwarf dogwoods, and (5) the big-bracted dogwoods were distinguished within *Cornus* [1,2]. The cornelian cherries are particularly used in traditional medicine and considered potential anti-diabetic and hypolipemic agents [3]. However, the knowledge on the main phytochemicals and biological activity of others dogwoods has been limited to date. The data on the secondary metabolites in the genus of *Cornus* indicate that mevalonic acid-derived iridoid glucosides, such as cornin, monotropein, and secologanin, occur in the red-fruited dogwoods in contrary to the blue-fruited ones. Alternatively, the blue-fruited dogwoods contain the phenolic glucosides like salidroside which are derived from shikimic acid [4,5]. In the extracts of better known cornelian cherries polyphenolic compounds (e.g., flavonoids and anthocyanins) are present. These chemicals are believed to exert anti-diabetic effect through inhibition of *α*-glucosidase activity combined with decreasing postprandial blood glucose level [6,7,8]. Some medicinal plants, including these belonging to Cornaceae, and their polyphenolic constituents are considered potential antioxidants and inhibitors of metabolic disorders complications [9,10,11]. It is believed that oxidative stress accelerate the development of serious symptoms of diseases such as diabetes mellitus [12]. Our previous study highlighted to some extent a role of flavonoids detected in the extracts from fruits of *Cornus alba* L. in the inhibition of digestive enzymes [13]. The compounds identified in the CA fruit extract were quercetin-3-*O*-glucuronide, kaempferol and quercetin 3-*O*-glucosides, as well as hydrolysable tannins, including 1,2,3,4,6-penta-*O*-galloylglucoside (PGG), and cornusiins A and B [13,14].

However, the phytochemical analysis of dogwood species, in particular less known ones, is justified. Therefore, fruits of representative species of opposite-leaved clade, such as *C. alba* (CA) and *C. sanguinea* (CS) subsp. *hungarica*, as well as big-bracted clade, such as *C. florida* (CF), have been selected for the investigation (Appendix A). Their potential biological activity, in particular an effect on digestive enzymes as well as their antioxidative ability which are linked with metabolic disorders, seem to be worthy of study in order to compare their activity with species more often used in a diet and to determine their alternative usage.

Thus, the aim of the study was to analyze the composition of aqueous-ethanolic extracts prepared from fruits of these species. Searching the biologically active compounds, the isolation procedures were included in the study. To compare the biological activity of extracts prepared from fruits of CA, CF and CS, their ability to scavenge 2,2-diphenyl-1-picrylhydrazyl (DPPH) as well as to inhibit chosen digestive enzymes, namely *α*-glucosidase (E.C. 3.2.1.20), *α*-amylase (EC 3.2.1.1) and pancreatic lipase (EC 3.1.1.3), was determined.

## 2. Results

### 2.1. Total Phenols Content

The quantitative analysis of the extracts was conducted in order to establish the total polyphenols content in the water-ethanol extracts from CF and CS. The content of phenolic compounds expressed as gallic acid equivalents in extract from fruits of CF was 24.30 ± 1.72 g/mg of dry weight (* *p* < 0.05 vs. *Cornus sanguinea*). The content of these compounds in extract from fruits of CS (19.72 ± 1.86 g/mg) turned out to be significantly lower. The total content of phenolic compounds in the aqueous-ethanolic extract from the fruits of CA was previously determined [13].

### 2.2. Phytochemical Analysis of Extracts

The qualitative analysis with HPLC-DAD-MS/MS allowed to detect 18 compounds in the CA fruit extract, 18 compounds in the CS fruit extract and 16 compounds in the CF fruit extract which belong to phytochemical classes such as iridoids and flavonoids (Figure 1, Figure 2, Figure 3).

In the extract from fruits of CA a significant amount of flavonoid glycosides (Figure 1, Table 1) was found such as quercetin-3-*O*-galactoside (**CA10**, R_t_ = 36.2, [M-H]^−^
*m*/*z* = 463), quercetin 3-*O*-glucuronide (**CA12**, R_t_ = 38.0, [M-H]^−^
*m*/*z* = 477), kaempferol 3-*O*-glucoside (**CA15**, R_t_ = 41.6, [M-H]^−^
*m*/*z* = 447.0) and kaempferol 3-*O*-glucuronide (**CA16**, R_t_ = 42.5, [M-H]^−^
*m*/*z* = 461). Few kaempferol hexosides, as well as their malonyl-derivatives, were annotated based on the MS/MS fragmentation pattern *m*/*z* 285 of aglycon after the sugar and malonyl moiety loss at R_t_ = 39.7 (**CA13**, [M-H]^−^ at *m*/*z* = 447), 44.6 (**CA17**, [M-H]^−^ at *m*/*z* = 533) and 46.4 (**CA18**, [M-H]^−^ at *m*/*z* = 533). In addition, although the anthocyanins peak were not observed in the UV chromatogram, the characteristic signals of anthocyanin patterns were noted in the extracted ion chromatograms at *m*/*z* 449, *m*/*z* 465 and *m*/*z* 611 in positive ESI mode. The ions [M+H]^+^ (*m*/*z* 449) in a positive ESI mode have been in the CA extract at R_t_ = 39.7 min. and R_t_ = 41.5 min. The major MS^2^ fragmentation pattern of these ions showed a signal at *m*/*z* 287. Thus, the compounds were identified as cyanidin hexosides. Moreover, additional signals at [M+H]^+^ (*m*/*z* 465) were registered at R_t_ = 36.1 min. and R_t_ = 37.0 min. in the extracted ESI positive ion chromatogram. The major MS^2^ fragmentation pattern of these ions showed a signal at *m*/*z* 303 which suggests the presence of delphinidin hexosides. The [M+H]^+^ signal of a compound (R_t_ = 35.2 min.) at *m*/*z* 611 and its MS^2^ fragmentation pattern at *m*/*z* 465 and *m*/*z* 303 in the positive ESI mode was tentatively assigned to delphinidin rhamnohexoside.

On the other hand, the most abundant compounds detected in the CS fruit extract were **CS4 (**R_t_ = 18.6, [M-H]^−^
*m*/*z* = 787), **CS11** (R_t_ = 30.0, [M-H]^−^
*m*/*z* = 625) and **CS17** (R_t_ = 38.0, [M-H]^−^
*m*/*z* = 477). All compounds showed diagnostic fragment ion at *m*/*z* 301 yielded after the loss of sugar moieties. The MS/MS fragment ions generated by compound **CS4** were *m*/*z* 625, *m*/*z* 463, *m*/*z* 301. Compound **CS11** ([M-H]^−^ at *m*/*z* = 625) seems to be a derivative of compound **CS4** ([M-H]^−^ at *m*/*z* 787) which could result from the loss of hexosyl fragment [M-hexosyl-H]^−^of the ion precursor *m*/*z* 787. Therefore, these compounds were identified as quercetin polyglycosides, whereas compound **CS17** was assigned as quercetin-3-*O*-glucuronide. In addition to quercetin, the polyglycosides of myricetin (**CS9**, **CS12**, **CS13**) and isorhamnetin (**CS8**, **CS13**) have been tentatively assigned in CS extract (Figure 2, Table 2).

In the aqueous-ethanolic extracts from the CF fruits, compounds from the iridoid as well as flavonoid groups have been identified (Figure 3, Table 3). The most abundant compounds have been detected at the retention times 6.3 min. (**CF1**), 22.0 min. (**CF4**), 36.1 min. (**CF11**) and 37.0 min. (**CF12**). The MS^2^ fragmentation pattern of compound **CF4** (*m*/*z* 433) showed signal at *m*/*z* 387, 225, 179 in negative ionization mode. Thus, compound **CF4** has been tentatively identified as dehydrologanin. Another iridoid, loganic acid (**CF2**, [M-H]^−^ at *m*/*z* 375) was detected in the CF extract. The MS/MS fragmentation patterns at *m*/*z* 213 [M-glucosyl-H]^−^ and *m*/*z* 169 [aglycon-CO_2_-H]^−^ were similar to the recently described in *Cornus* genus [15]. The [M-H]^−^ peaks of quercetin 3-*O*-galactoside (**CF11**, R_t_ = 36.1 min.) and quercetin 3-*O*-glucoside (**CF12**, R_t_ = 37.0 min.) at *m*/*z* 463 were detected in the negative ESI mode. The MS^2^ fragmentation pattern of compounds **CF11** and **CF12** showed a signal at *m*/*z* 301 indicating the presence of sugar moiety in the deprotonated ion [M-H]^−^ at *m*/*z* 463. The major ion in MS spectrum of compounds **CF1** was [M-H]^−^
*m*/*z* 345 in negative ESI mode. The MS/MS spectrum of the precursor ion *m*/*z* 345 (**CF1)** exhibited main fragment ion at *m*/*z* 169 that led to the tentative identification of compound **CF1** as glucuronide of gallic acid.

The isolation of compounds from the acetone–methanol–water extracts from CA fruits was carried out according to scheme displayed in the Appendix A. Nine compounds were isolated from the subfractions D and H by column chromatography and preparative HPLC among which seven had their structure fully elucidated. Compound D1.1 showed the peak at *m*/*z* = 337 in the negative ESI mode. In addition, based on the retention time and MS/MS spectra, it was found that compounds D1.1 and D2.3 are identical. The comparison of NMR spectra with those available in the literature compounds D1.1 and D2.3 were identified as 5-*O* -coumaroylquinic acid [16]. Similarly, the data were the same for compounds D1.2 and D2.4. Thus, they were identified as 5-*O*-(*E*)-*p*-coumaroylquinic acid methyl ester ([M-H]^−^ at *m*/*z* = 351) [17]. Compound D2.1 was identified as chlorogenic acid by comparing the retention time with a standard and NMR data available in the literature [18]. The structure of compound D2.5 was found to be kaempferol 3-*O*-glucuronide 6″-methyl ester ([M-H]^−^
*m*/*z* = 475) [19]. Last but not least, compound H1.1 (**CA3**) was identified as hydroxytyrosol glucoside ([M-H]^−^ at *m*/*z* = 315) based on the data previously described [13,20]. The NMR spectra of identified compounds were provided in the Appendix A.

Initially, the purpose was to isolate the most abundant compounds of the extract acetone – methanol – water extracts from CA fruits which are supposed to determine its activity. However, due to limited separation of flavonoids only minor compounds, which have not been detected in the aqueous-ethanolic extract of CA fruit, except for compound **CA3** (corresponding to the isolated compound H1.1), or artefacts such as methyl derivatives of 5-*O*-(*E*)-*p*-coumaroylquinic acid or kaempferol 3-*O*-glucuronide generated in the extraction procedures, were isolated.

### 2.3. Antioxidant Activity

The antioxidant activity of extracts was assayed by the determination of the DPPH (1,1-diphenyl-picrylhydrazyl radical) scavenging ability (Table 4). The lowest half scavenging concentration (SC_50_) value, and thus the highest antioxidant activity, was shown by the aqueous-ethanolic extract from the fruits of CF (17.10 ± 6.89 μg/mL). The statistically significant difference between scavenging activities of CF and ascorbic acid has not been noted. The SC_50_ value of CA fruit extract (91.47 ± 3.66 μg/mL) and the SC_50_ value for CS fruits extract (130.03 ± 31.61 μg/mL) were significantly higher than SC_50_ of CF extract. There was no significant difference between activities of CA and CS extracts. In addition, the CA and CS extracts were less active (*p* < 0.001) than ascorbic acid (3.73 ± 0.83 μg/mL).

### 2.4. Digestive Enzymes Inhibition

The inhibition of digestive enzymes was studied by the fluorimetric and spectrophotometric methods. The results are shown in Table 5. Firstly, the inhibition of *α*-glucosidase activity was assessed spectrophotometrically. The highest activity was determined for the aqueous-ethanolic extract from fruits of CF, (IC_50_ = 38.87 ± 2.65 μg/mL). The IC_50_ values of the aqueous-ethanolic extracts from fruits of CA and CS were 50.38 ± 14.05 μg/mL and 70.07 ± 16.62 μg/mL, respectively. It should be underlined that the standard inhibitor of *α*-glucosidase acarbose (IC_50_ = 150.43 ± 0.04 μg/mL; 233.0 ± 0.06 µM) was significantly less active than tested in this assay extracts of CA fruits (*p* < 0.05) and CF fruits (*p* < 0.001).

Secondly, in order to further assess the potential role of dogwood extracts in the regulation of polysaccharides digestion, their inhibitory activity towards *α*-amylase was studied. Based on the results of the experiments, it was found that the aqueous-ethanolic extract from the fruits of CA was the most active with the IC_50_ value 115.20 ± 14.31 μg/mL. The IC_50_ values of the aqueous-ethanolic extracts from fruits of CS and CF were determined for the first time and equaled 651.44 ± 12.99 μg/mL and 5018.43 ± 14.70 μg/mL, respectively (Table 5).

Thirdly, the PL inhibition by the dogwood extracts was determined. Among all tested extracts, the only inhibitory PL activity was found in the case of the aqueous-ethanolic extract from fruits of CA, whereas the extracts from fruits of CF and CS were inactive (data not shown). However, all tested extracts were less active than the PL inhibitor orlistat (1.49 ± 0.2 ng/mL; 3.0 ± 0.4 nM) and *α*-amylase inhibitor acarbose (2.4 ± 0.06 µg/mL; 3.7 ± 0.6 µM; *p* < 0.001). It is worth to note that all tested extracts prepared from dogwood fruits were more active inhibitors of *α*-glucosidase activity compared to *α*-amylase and PL activity.

Taking into consideration all the results obtained both in this and our previous study [13], it was concluded that the aqueous-ethanolic extract from fruits of CA was the most active one. In order to find the phytochemicals responsible for this activity, the isolation of compounds from acetone-methanol-water extract (3:1:1, *v*/*v*/*v*) was carried out. Prior to the tests, the ability of this extract to inhibit the activity of digestive enzymes was determined. The most relevant activity of acetone-methanol-water extract from fruits of CA, was determined in the case of PL (IC_50_ = 16.34 ± 4.51 μg/mL). The inhibition of *α*-glucosidase activity was found to be less significant (IC_50_ = 34.75 ± 7.52 μg/mL). The IC_50_ value for inhibition of *α*-amylase was 58.60 ± 11.18 μg/mL. The acetone-methanol-water extract of CA fruits had higher activity than the aqueous-ethanolic extract of CA fruits. The isolated constituents of the acetone-methanol-water extract were further investigated in the *α*-glucosidase activity assay (Table 6). The most active inhibitors of *α*-glucosidase were 5-*O*-(*E*)-*p*-coumaroylquinic acid methyl ester (D2.4) and kaempferol 3-*O*-glucuronide 6″-methyl-ester (D2.5). The IC_50_ values were 0.16 ± 0.01 mM and 0.21 ± 0.05 mM, respectively.

## 3. Discussion

Plants are the most important sources of chemical compounds which often exert pleiotropic effect that plays a key role in metabolic disorders. Presence of pancreatic enzymes (lipase, amylase) inhibitors in natural resources has been reported. Traditionally, some herbs have been used directly for treatment of metabolic disorders. On the other hand, more and more clinically effective substances might be obtained from plants even though they are not classified as medicinal plants. Nowadays, medicinal plants due to their simple accessibility and fewer adverse effects provide the potentially active compounds which are likely to play a role in the treatment of some disorders [21]. It seems that some classes of secondary metabolites derived from plant materials, such as tannins, flavonoids, xanthones, procyanidins and caffeoylquinic acid derivatives, are active inhibitors of *α*-glucosidase, *α*-amylase and PL [21,22]. A wide range of fruits, including cherries rich in phenolic compounds, has been used due to their dietary attributes and for this reason they are a precious material for industry. Fruits of *Cornus mas* L. (cornelian cherries) have been applied in the food industry for the production of tinctures, juices or confitures in Europe, whereas fruits of *Macropcarpium officinale* (*Cornus officinale*) Sieb. and Zucc. are used in the traditional Chinese medicine in addition to industrial application [23]. On the other hand, many species of *Cornus* genus play a decorative role in the industrial environment. Thus, a few of them were used in this study in order to describe their chemical composition as well as antioxidant and anti-digestive enzymes activity which have not been widely studied according to our knowledge.

Firstly, the total phenols content was determined. Among all studied extracts, the highest content of phenolic compounds was found in the aqueous-ethanolic extract from the fruits of CA (39.60 ± 2.28 mg GAE/g of extract) as it was described in our previous study [13]. The content of phenolic compounds expressed as gallic acid equivalent in extract from fruits of CF was determined for the first time (24.30 ± 1.72 mg GAE/g). It was significantly higher than in the extract from fruits of CS subsp. *hungarica* (19.72 ± 1.86 mg GAE/g). The determined TPC of CS fruit extract has been comparable to that previously described in the ethyl acetate extract from fruits of *Cornus sanguinea* L. subsp. *australis*, 20.13 ± 0.02 mg/g of dry extract. However, in the same study TPC of the aqueous and methanolic extract was 74.91 ± 0.01 mg/g and 88.56 ± 0.04 mg/g of dry extract, respectively [24]. On the other hand, Stanković and Topuzović (2012) also determined TPC in the methanolic (34.19 ± 0.25 mg GAE/g), aqueous (27.45 ± 0.15 mg GAE/g) and ethyl acetate (45.34 ± 0.19 mg GAE/g) extracts of CS fruits [25]. It was reported that most of phenolic secondary metabolites including flavonoids of CS have been mainly concentrated in leaves [25].

Following the TPC determination, which is directly linked with antioxidant properties, the DPPH assay was used for the preliminary investigation of antioxidant activity of dogwoods extracts. The obtained results for CS (SC_50_ = 130.03 ± 31.61 µg/mL) were significantly lower than previously reported SC_50_ of ethyl acetate, aqueous and methanolic of CS subsp. *australis* fruit extracts (90.43–762.3 mg/mL) [24]. In another study the CS antioxidant activity was also less relevant, the IC_50_ values determined through DPPH scavenging activity were 358.59 ± 1.14 µg/mL, 384.45 ± 2.01 µg/mL and 537.83 ± 1.98 µg/mL for methanolic, aqueous and ethyl acetate extract of CS fruit, respectively [25]. On the other hand, the lowest SC_50_ value, and thus the highest antioxidant activity, was shown by the aqueous-ethanolic extract from the fruits of CF (17.10 ± 6.89 μg/mL) which was comparable with ascorbic acid. Therefore, it seems that phytochemicals of aqueous-ethanolic CF extract have exerted more relevant radical scavenging activity than others. According to our knowledge, it has been determined for the first time in this study. It should be emphasised that in this extract mainly iridoids have been identified. The iridoid-type compounds are believed to exert less relevant antioxidant activity than phenolic type compounds such as flavonoids and anthocyanins. It was proved that a fractions (1 mg/mL) of *Corni fructus* containing iridoids like loganin (content 83.07%) and morroniside (content 16.93%) or morroniside alone are less active than the fraction containing 5-hydroxymethylfurfural (content 21.10%), morroniside (content 5.40%) and loganin (content 64.05%) enriched in gallic acid (content 9.45%) [26]. Thus, it is suggested that minor compounds, other than iridoids, determine the antioxidant activity of CF extract.

According to the available data, *Cornus* spp. have been indicated as a rich source of phenolic compounds such as anthocyanins, including 3-galactosides and 3-glucosides of pelargonidin, cyanidin and delphinidin [27,28,29,30,31]. In addition, the representative compounds of this class have been detected both in red bark and in the berries of CA “Sibirica” [32]. *Cornus* anthocyanins have been established the anti-tumour, anti-inflammatory, and antioxidant compounds, and their role in the treatment of diabetes mellitus-related disorders has been considered [11,31,33]. The *Cornus* spp. anthocyanins such as delphinidin 3-*O*-*β*-galactoside, cyanidin 3-*O*-*β*-galactoside and pelargonidin 3-*O*-*β*-galactoside at concentration of 40 µM, showed antioxidant activities of 70.2, 60.1, and 40.3%, respectively, in an iron-catalyzed liposomal model [33]. Therefore, analyzing the phytochemical composition with HPLC-DAD-MS/MS we particularly considered the presence of anthocyanins apart from iridoids and flavonoids. In fact, the total anthocyanin content in the CA fruit ethanolic extract (4.0 ± 0.1 mg/g) was established in our previous study [13]. Taking into consideration that the signals of anthocyanins were detected only in the extracted ion chromatograms in positive ESI mode, the trace amounts of anthocyanins in the aqueous-ethanolic extract were found in this study. Our investigation have also purposed to isolate the compounds from CA fruit extract to determine their activity. The acetone–methanol–water extract from CA fruits was used for isolation procedures. Prior the isolation its activity was studied in PL, *α*-amylase and *α*-glucosidase assays. The acetone-methano-water extract was even more active than the aqueous-ethanolic extract of CA fruits. Although flavonoids have been particularly detected in both extracts, minor compounds such as chlorogenic acid (1) and 5-*O*-coumaroylquinic acid (2) were finally isolated due to the limited separation of flavonoid compounds primary detected in the CA extracts. The supposed low quantities of these phenolic acids limit their detection in the crude extracts. On the other hand, two of the isolated chemicals including 5-*O*-(*E*)-*p*-coumaroylquinic acid methyl ester (3) and kaempferol 3-*O*-glucuronide 6″-methyl-ester (4) seem to be artefacts. It is concluded that the higher activity of the acetone-methanol-water extract is a result of the presence of methylated derivatives which probably exert more relevant inhibitory activity against digestive enzymes.

The inhibition of digestive enzymes studies showed that the aqueous-ethanolic extract from the fruits of CA was the most active *α*-amylase inhibitor and the only PL inhibitor (5.61 ± 1.72 mg/mL). The extracts from fruits of CF and CS were completely inactive in the PL assay. On the other hand, the ability to inhibit *α*-glucosidase activity by extracts prepared from fruits of CA, CS and CF was determined for the first time in this study. Based on the preliminary results concerning the activity of CA preparations towards digestive enzymes, CA fruits have been subsequently investigated. In fact, the most abundant chemicals detected in the CA aqueous-ethanolic extract are the flavonols such as quercetin 3-*O*-galactoside (**CA10**), quercetin 3-*O*-glucuronide (**CA12**) as well as kaempferol 3-*O*-glucoside (**CA15**) and kaempferol 3-*O*-glucuronide (**CA16**). According to the available literature, flavonoids including quercetin and kaempferol glycosides are able to inhibit the activity of digestive enzymes [34]. It was shown that flavonoids and phenolic acids are stronger inhibitors of *α*-amylase and *α*-glucosidase than PL activity. In the *α*-glucosidase assay, quercetin-3-*O*-glucoside (IC_50_ = 12.34 µM), kaempferol-3-*O*-rutinoside (IC_50_ = 10.18 µM) and kaempferol-3-*O*-glucoside (IC_50_ = 15.90 µM) showed significantly higher activity than gallic acid (IC_50_ = 296.20 µM), caffeic acid (IC_50_ = 223.30 µM), chlorogenic acid (IC_50_ = 231.80 µM) or ferulic acid (IC_50_ = 460.76 µM) [34]. Few compounds, such as rhododendrin, emodin and physcion, which have been isolated from roots of *Rheum turkestanicum* showed a relevant anti-glucosidase activity (Dehghan et al., 2018). Moreover, the IC_50_ against PL activity of quercetin-3-*O*-glucoside, kaempferol-3-*O*-glucoside as well as gallic acid and chlorogenic acid were as follows 1.74 mM, 3.52 mM, 5.05 mM and 3.81 mM, respectively [34]. These data support our conclusion that flavonoids and phenolic acids inhibit PL activity in the significantly higher concentrations, which might explain a low anti-PL activity of CA extract and lack of activity of CS and CF extracts. The phenolic compounds isolated from fruit of CA exerted rather weak anti-glucosidase activity in our study. The IC_50_ = 0.80 ± 0.32 mM, determined for chlorogenic acid (**1**, D2.1), was less relevant than previously described by Tan and Chang (2017). Additionally, it is suggested that methylated derivatives more significantly inhibited *α*-glucosidase activity than other phenolic compounds. However, it should be noted that these compounds might be the artefacts.

## 4. Materials and Methods

### 4.1. General Experimental Procedures

Acetonitrile (MeCN, UHPLC-grade), *n*-butanol (BuOH), chloroform (CHCl_3_), ethyl acetate (EtOAc), ethanol (EtOH) and methanol (MeOH) for extraction were obtained from POCH (Gliwice, Poland). Natural product reagent A (diphenylboric acid 2-aminoethyl ester) and quercetin 3-*O*-glucoside (isoquercitrin) were purchased from Carl Roth GmbH (Karlsruhe, Germany)**.** Formic acid (HCOOH) for uses as an additive in the UHPLC-MS eluent, orlistat (O4139-25MG), 4-methylumbelliferyl oleate (MUO, 75164-100MG), acarbose (A8980-1G) and pancreatin from porcine pancreas (P1750-100G) as well as *p*-nitrophenyl-*α*-d-glucopyranoside (N1377-1G) and *α*-glucosidase from *Saccharomyces cerevisiae* (G5003-100UN) were purchased from Sigma-Aldrich Chemie GmbH (Steinheim, Germany). The standards of kaempferol 3-*O*-glucoside (astragalin), quercetin 3-*O*-rhamnoside (quercitrin) and quercetin 3-*O*-rutinoside (rutin) were purchased from Sigma-Aldrich Chemie GmbH (Steinheim, Germany). Quercetin 3-*O*-galactoside (hyperoside) was purchased from HWI Analytik GmbH (Rheinzaberner, Germany). Chlorogenic acid was isolated in the Department of Pharmacognosy and Molecular Basis of Phytotherapy (Medical University of Warsaw, Poland) from herb of *Lamium album* L. The EnzChek™ Ultra Amylase Assay Kit (Invitrogen, Pailey, UK) was used in *α*-amylase inhibitory assay. Tris-HCl buffer was prepared using 13 mM Tris-HCl (Promega Corporation, Madison, USA), 150 mM NaCl (POCH, Gliwice, Poland), and 1.3 mM CaCl_2_ (POCH, Gliwice, Poland). Water was obtained using Millipore Simfilter Simplicity UV (Molsheim, France) water purification system. HPLC-DAD-MS^n^ analysis was performed on a UHPLC-3000 RS system (Dionex, Germany) with DAD and an AmaZon SL ion trap mass spectrometer with an ESI interface (Bruker Daltonik GmbH, Germany). Structures of the isolated compounds have been determined based on UV–Vis, MS, and ^1^H and ^13^C spectra. If needed, additional experiments, such as HSQC and HMBC were performed. The NMR spectra were acquired using Varian VNMRS 300 Oxford 300 MHz spectrometer (Bruker, Billerica, MA, USA).

### 4.2. Plant Material and Extracts Preparation

*Cornus alba* fruits were collected in the Ursynów District of Warsaw, Poland (52°09′01″ N, 21°03′01″ E) in September 2017. *Cornus florida* fruits were collected in September 2017 in the Botanical Garden—Center for Biological Diversity Conservation in Powsin (Polish Academy of Sciences, Poland) (52°06′17″ N, 21°05′42″ E). *Cornus sanguinea* subsp. *hungarica* fruits were collected in the Ursynów District of Warsaw, Poland (52°08′29″ N, 21°03′13″ E). Voucher specimens (No FW25_20160929_CA and FW25_20170525_CA, No FW25_20170821_CS, No FW25_20170919_CF) are available in the herbarium of the Department of Pharmacognosy and Molecular Basis of Phytotherapy, Medical University of Warsaw. The plant material was identified by Iwona Stanisławska and Konrad Woliński according to plant guidebooks [35].

A 10-g portion of powdered plant material was macerated four times with aqueous ethanol (60%, *v*/*v*) in a ratio of 1:10 for 24 h each time. The collected ethanolic extracts were concentrated under reduced pressure and lyophilized. The obtained dry weights of the ethanolic extracts from the fruit samples were 1.77 g (CA), 1.44 g (CF) and 1.44 g (CS).

### 4.3. Phytochemical Analysis

#### 4.3.1. HPLC-DAD-MS/MS

Separations were performed on a Kinetex XB-C_18_ column (150 × 2.1 mm, 1.7 μm) (Phenomenex, Torrance, USA). The column temperature was 25 °C. For preliminary phytochemical analyses of the extracts and fractions, mobile phase A was 0.1% HCOOH in water and mobile phase B was 0.1% HCOOH in acetonitrile. The gradient program was as follows: 0–50 min. 5–26% B; 50–60 min. 26–95% B. The flow rate was 0.2 mL/min. The column was equilibrated for 10 min between injections. UV spectra were recorded 200–800 nm range, and chromatograms were acquired at 240, 280, 325 nm or 520 nm. The LC eluate was introduced directly into the ESI interface without splitting. The nebulizer pressure was 40 psi; dry gas flow was 9 L/min; the oven temperature was 300 °C; and capillary voltage was 4.5 kV. Analyses were carried out scanning from *m*/*z* 200 to 2200. Compounds were analyzed in negative and positive ion mode. The MS^2^ fragmentation patterns were obtained for the most abundant ion.

#### 4.3.2. Total Polyphenols Content

The sum of phenols was determined using a modified spectrophotometric method with Folin-Ciocalteu’s reagent [36]. The assay was performed as previously described [37]. The absorbance was measured at 765 nm in a microplate reader (SYNERGY 4, BioTek, Winooski, VT, USA) and the results were expressed as gallic acid equivalent (mg of GAE/g of extract).

#### 4.3.3. Isolation of Compounds from CA Fruits

A 1619-g portion of powdered CA fruits was extracted four times in an ultrasonic bath for 1 h followed by the maceration with acetone-methanol-water (3:1:1, *v*/*v*/*v*) for next 23 h each time. The organic solvent was evaporated under reduced pressure at 40 °C and concentrated to the volume of 1 L. Next, the residue was partitioned between chloroform, ethyl acetate and *n*-butanol saturated with water, and each extraction was conducted with 1:1 ratio of residue and solvent (Appendix A).

The selected fraction EtOAc was subjected to column chromatography on silica gel (6.5 cm × 25 cm), eluted with solvents gradient (from 100% CHCl_3_ to 100% EtOAc, every 5%; from 100% EtOAc to 100% MeOH, every 10%) to give 230 fractions. These fractions were combined into 9 (A-I) main subfractions based on the TLC profiles (stationary phase, silica gel; mobile phase, EtOAc:HCOOH:acetic acid:water, 100:11:11:26, *v*/*v*/*v*/*v*) after derivatization with 1% Natural product reagent A. The subfractions D and H were subjected to column chromatography on Sephadex LH-20 (2 cm × 130 cm) and Toyopearl (4 cm × 40 cm), respectively, next eluted with MeOH (100%) and MeOH (50%, *v*/*v*), respectively to give subfractions D1-D5 and H1-H5. The subfractions D1, D2 and H1 were subjected to preparative HPLC system (Shimadzu LC10vp, Kyoto, Japan, Kinetex XB-C_18_–5 µm, 150 mm × 21.2 mm, Agilent Technologies, Santa Clara, USA, 280 nm, flow 20 mL/min, mobile phase: 0.1% HCOOH in water (A) and 0.1% HCOOH in acetonitrile (B); elution program: 0% B–35% B (0–35 min). Fractions were collected based on UV-Vis chromatograms to give pure compounds D1.1 (17.3–18.0 min.; 8.75 mg), D1.2 (23.3–23.9 min.; 12.66 mg), D1.3 (24.4–24.8 min.; 6.66 mg) from the subfraction D1, compounds D2.1 (14.5–14.8 min.; 10 mg), D2.2 (16.0–16.3 min.), D2.3 (17.35–17.75 min.; 3.88 mg), D2.4 (23.2–23.6 min.; 5.63 mg) and D2.5 (28.0–28.5 min.; 12.69 mg) from the subfraction D2 and compound H1.1 (4.9–5.4 min.; 75.86 mg) from the subfraction H1. Isolated compounds were identified based on UV-Vis, MS and ^1^H and ^13^C NMR spectra.

### 4.4. Biological Experiments

#### 4.4.1. DPPH Assay

The scavenging of DPPH radical was examined using a modified assay previously described [38]. The equal volumes (100 µL) of ethanolic solution of DPPH (0.02 M) and extract samples dissolved in 50% (*v*/*v*) aqueous-ethanol were added into 96-well microplates and incubated for 30 min in the darkness. The absorbance was measured at 518 nm with microplate reader (Synergy 4, Biotek, USA). The blind samples were prepared in order to exclude color interference. Ascorbic acid was used as a positive control. The scavenging activities of extracts were displayed as SC_50_ values.

#### 4.4.2. *α*-Glucosidase Inhibition

The *α*-glucosidase inhibition by extracts was determined spectrophotometrically according to the modified assay previously described [39]. The colorimetric measurement of enzyme activity was based on the hydrolytic cleavage of a substrate, *p*-nitrophenyl-*α*-d-glucopyranoside (pNPG). Briefly, sample solution of tested extract or isolated compound (150 µl) dissolved in 0.1 M phosphate buffer (pH 6.8) was incubated with 50 µL of *α*-glucosidase solution (0.4 U/mL) at 37 °C. After 15 min of incubation, the substrate solution (0.7 mM in 0.1 M phosphate buffer, pH 6.8) was added and the second 15 min incubation at 37 °C was performed. The reaction was terminated by adding 50 µL sodium carbonate solution (0.2 M). The absorbance of *p*-nitrophenol was measured at 405 nm in a microplate reader (Synergy 4, Biotek, USA). Acarbose was used as a positive control. A control without test samples or acarbose represented 100% *α*-glucosidase activity. The *α*-glucosidase inhibitory activities of extracts were displayed as half inhibitory concentration (IC_50_) values.

#### 4.4.3. *α*-Amylase Inhibition

An EnzChek™ *Ultra Amylase Assay Kit* was used to determine the *α*-amylase activity as described previously [40]. The fluorescence method was based on the hydrolytic cleavage of a modified starch derivative (DQ^TM^ starch from corn, BODIPY^®^ FL conjugate, 200 µg/mL). Porcine pancreas powder (0.5 mg/mL in Tris-HCl buffer, pH 8.0) was used as the enzyme source. The tested extracts were dissolved in DMSO. The enzyme and substrate solutions, as well as the test samples, were prepared immediately before use. The fluorescence of the starch derivative was measured at excitation and emission wavelengths of 485 nm and 535 nm, respectively, at 37 °C in a microplate reader (Synergy 4, Biotek, USA). Acarbose was used as a positive control. A control without test samples or acarbose represented 100% *α*-amylase activity. The *α*-amylase inhibitory activities of extracts were displayed as half inhibitory concentration (IC_50_) values.

#### 4.4.4. Pancreatic Lipase Activity

A previously described enzymatic in vitro assay based on the hydrolysis kinetics of the oleate ester of 4-methylumbelliferone (0.5 mM) was used to determine the PL activity and inhibitory potential of the extracts [40]. Porcine pancreas powder (10 mg/mL in Tris-HCl buffer, pH 8.0) was used as the enzyme source. The tested extracts were dissolved in DMSO. The fluorescence of 4-methylumbelliferone was measured at excitation and emission wavelengths of 360 nm and 465 nm, respectively, at 37 °C in a microplate reader (Synergy 4, Biotek, USA). Orlistat was used as a positive control. A control without test extracts or orlistat represented 100% PL activity. The PL inhibitory activities of extracts were displayed as half inhibitory concentration (IC_50_) values.

### 4.5. Statistical Analysis

The results are expressed as a means ± SD. Each sample of extract, fraction or compound was tested in triplicate in three independent experiments. Statistical significance of the differences between means was established by testing homogeneity of variance and a normality of distribution followed by ANOVA with Tukey’s *post hoc* test or non-parametric methods such as Kruskal-Wallis test. The *P* values below 0.05 were considered statistically significant. All analyses were performed using Statistica 10 (StatSoft, Cracow, Poland).

## 5. Conclusions

In conclusion, plants are the source of potent free radical scavengers as well as digestive enzymes inhibitors. Therefore, the investigations of biological activity and chemical composition of non-feed species that are widespread in the rural area are justified. Our study considering the digestive enzymes inhibitory potential of *Cornus* spp. extracts as well as their antioxidant activity shows that flavonoids or phenolic acids more significantly than iridoids inhibit *α*-amylase, *α*-glucosidase, and PL activity. The isolated phenolic-type compounds of CA are likely to play a potential role in the postprandial glucose blood concentration via modulation of *α*-glucosidase activity. Thus, the significance of CA as a source of bioactive compounds should not be neglected. Last but not least, the HPLC-DAD-MS/MS analysis allowed to provide the data on less known red-fruited dogwood species, CF. Based on the phytochemical analysis, it was classified to the branch of dogwoods rich in the mevalonic acid-derived iridoid glucosides. In addition, the extract of this species was characterized by more significant content of phenolic compounds compared to other tested extracts and was the most relevant DPPH scavenger. Thus, the phytochemicals of CF seem to be particularly worthy of further investigation.

## Figures and Tables

**Figure 1 plants-09-00122-f001:**
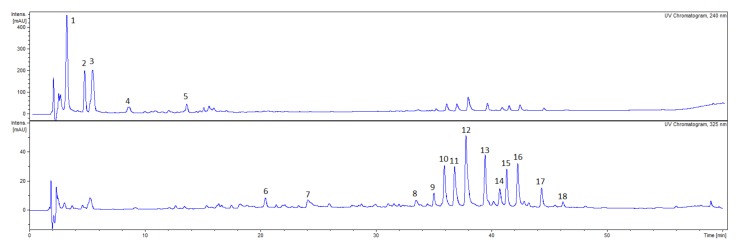
HPLC chromatograms of the ethanolic extracts from fruits of *Cornus alba* (CA) (10 mg/mL) acquired at 240 nm and 325 nm. HPLC conditions: Kinetex XB-C_18_ column (150 × 2.1 mm, 1.7 μm), mobile phase A: 0.1% HCOOH/H_2_O; B: 0.1% HCOOH/MeCN, and the gradient was as follows: 0–50 min. 5–26% B; 50–60 min. 26–95% B.

**Figure 2 plants-09-00122-f002:**
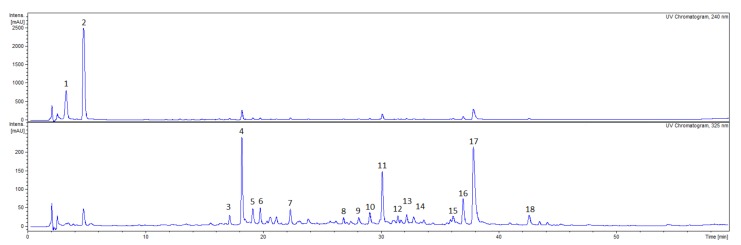
HPLC chromatograms of the ethanolic extracts from fruits of *Cornus sanguinea (*CS) (10 mg/mL) acquired at 240 nm and 325 nm. HPLC conditions: Kinetex XB-C_18_ column (150 × 2.1 mm, 1.7 μm), mobile phase A: 0.1% HCOOH/H_2_O; B: 0.1% HCOOH/MeCN, and the gradient was as follows: 0–50 min. 5–26% B; 50–60 min. 26–95% B.

**Figure 3 plants-09-00122-f003:**
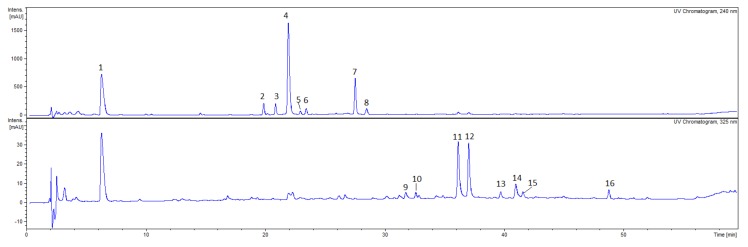
HPLC chromatograms of the ethanolic extracts from fruits of *Cornus florida* (CF) (10 mg/mL) acquired at 240 nm and 325 nm. HPLC conditions: Kinetex XB-C_18_ column (150 × 2.1 mm, 1.7 μm), mobile phase A: 0.1% HCOOH/H_2_O; B: 0.1% HCOOH/MeCN, and the gradient was as follows: 0–50 min. 5–26% B; 50–60 min. 26–95% B.

**Table 1 plants-09-00122-t001:** Chromatographic and spectrometric data of compounds identified in *Cornus alba* fruit extract.

No.	Tentative Assignement	Rt [min]	UV [nm]	[M-H]^−^ *m*/*z*	MS^2^ Ions *m*/*z*
**CA1**	Unidentified	3.2	226	491	431, 373
**CA2**	Unidentified	4.8	217	689	527, 515, 481, 353, 172
**CA3**	Hydroxytyrosol glucoside	5.3	290	631 *	315
**CA4**	Unidentified	8.5	219	665	491, 373
**CA5**	Unidentified	13.6	290	489	314, 173
**CA6**	Unidentified	20.6	290	663	616, 521, 489, 405, 329
**CA7**	Coumaroylquinic acid Tetragalloylglucose	24.3 24.3	290, 310sh overlapped	337 787	191, 163, 119 453, 315
**CA8**	Quercetin-3-*O*-(2″-*O*-galloyl)-hexoside	33.7	270, 360	615	463, 301
**CA9**	Quercetin 3-*O*-rhamnoglucoside	35.2	270, 350	609	463, 301
**CA10**	Quercetin 3-*O*-galactoside	36.2	260, 353	463	301
**CA11**	Kaempferol derivative Quercetin hexoside	37.0 overlapped	260, 352	599 463	447, 313, 285, 169 301
**CA12**	Quercetin 3-*O*-glucuronide	38.0	255, 352	477	301
**CA13**	Kaempferol hexoside	39.7	264, 349	447	285
**CA14**	Quercetin (6″-*O*-malonyl)-3-*O*-*β*-d-glucoside	41.2	265, 353	549	505, 463, 343, 301
**CA15**	Kaempferol 3-*O*-glucoside	41.6	264, 344	447	327, 285
**CA16**	Kaempferol 3-*O*-glucuronide	42.5	264, 344	461	285, 175
**CA17**	Kaempferol malonyl-hexoside	44.6	264, 344	533	489, 447, 285
**CA18**	Kaempferol malonyl-hexoside	46.4	264, 350	533	489, 285

* adduct.

**Table 2 plants-09-00122-t002:** Chromatographic and spectrometric data of compounds identified in *Cornus sanguinea* fruit extract.

No.	Tentative Assignment	Rt [min]	UV [nm]	[M-H]^−^ *m*/*z*	MS^2^ *m*/*z*
**CS1**	Unidentified	3.1	230	631	315, 243, 161
**CS2**	Unidentified	4.8	236, 315	523	505, 477, 387, 315, 232, 179
**CS3**	Quercetin tri-hexoside	17.1	262, 340	787	625, 463, 301
**CS4**	Quercetin tri-hexoside	18.2	262, 342	787	625, 463, 301
**CS5**	Myricetin tri-hexoside Quercetin-dipentosyl-hexoside	19.0 overlapped	263, 342	803 757	639, 479, 317 625, 595, 463, 301
**CS6**	Myricetin tri-hexoside	19.9	264, 330	803	641, 479, 317
**CS7**	Myricetin tri-hexoside Quercetin dihexoside	22.3 overlapped	256, 360	803 625	641, 479, 317 463, 301
**CS8**	Isorhamnetin tri-hexoside	26.8	270, 350	639	477, 315
**CS9**	Myricetin dihexoside Quercetin dihexoside	28.0 overlapped	250, 360	641 625	479, 317 463, 301
**CS10**	Quercetin dihexoside	29.0	270, 360	625	463, 301
**CS11**	Myricetin dihexoside Quercetin dihexoside	30.0 overlapped	264, 342	641 625	479, 317 463, 301
**CS12**	Myrcetin dihexoside	31.4	270, 340	641	479, 317, 303
**CS13**	Myricetin dihexoside Isorhamnetin tri-hexoside	32.2 overlapped	260, 340	641 639	479, 317, 303 477, 396, 272
**CS14**	Quercetin dihexoside Quercetin 6-*O*-*α*-l-arabinosyl-d-glucoside	32.8 overlapped	260, 350	625 595	463, 301 475, 463, 449, 300, 179
**CS15**	Quercetin 3-*O*-rhamnoglucoside	36.1	260, 352	609	301
**CS16**	Quercetin hexoside	37.0	255, 353	463	301
**CS17**	Quercetin 3-*O*-glucuronide	38.0	255, 353	477	301
**CS18**	Quercetin hexoside	42.8	260, 350	463	301

**Table 3 plants-09-00122-t003:** Chromatographic and spectrometric data of compounds identified in *Cornus florida* fruit extract.

No.	Tentative Assignment	Rt [min]	UV [nm]	[M-H]^−^ *m*/*z*	MS^2^ Ions *m*/*z*
**CF1**	Gallic acid glucuronide	6.3	254, 297	345	169, 123
**CF2**	Loganic acid	19.9	233	375	213, 169
**CF3**	Unidentified	20.9	236	595 ^#^	549, 341, 281, 225
**CF4**	Dehydrologanin derivative	22.0	238	433 ^#^	387, 225, 179
**CF5**	Dehydrologanin derivative	22.9	223, 280	433 ^#^	387, 225, 179, 149
**CF6**	Loganin	23.4	234, 280	435 ^#^	389, 227
**CF7**	7-*epi*-loganin	27.4	232	435 ^#^	389, 227
**CF8**	Unidentified	28.5	235	539	491, 343, 195
**CF9**	Quercetin-hexose protocatechuic acid Myricetin hexoside	31.3 overlapped	280, 350sh	599 479	483, 465, 405, 301 317, 179
**CF10**	Quercetin 6-*O*-*α*-l-arabinosyl-d-glucoside	32.5	275, 360	595	463, 301, 179
**CF11**	Tetragalloylglucose Quercetin 3-*O*-galactoside	36.1 overlapped	260, 360	787 463	635, 465, 319, 249 301
**CF12**	Quercetin hexoside	37.0	260, 352	463	301
**CF13**	Tetragalloylglucose Kaempferol hexoside	39.7 overlapped	265, 353	787 447	633, 573, 465, 379, 249 327, 284, 151
**CF14**	Quercetin (6″-*O*-malonyl)-3-*O*-*β*-d-glucoside	41.2	280, 360	549	505
**CF15**	Unidentified Kaempferol hexoside	41.5 overlapped	280	615 447	567, 419, 223, 195, 177 285
**CF16**	Unidentified Unidentified	48.8 overlapped	280	533 712	307, 225, 163 666, 605, 504, 442, 377, 322

^#^ [M-H+HCOOH]^−^.

**Table 4 plants-09-00122-t004:** The SC_50_ values (µg/mL) of the extracts from fruits of *Cornus alba*, *Cornus florida*, and *Cornus sanguinea* for the scavenging of DPPH.

*Aqueous-Ethanolic Extract*	DPPH Scavenging (µg/mL)
***Cornus alba***	91.47 ± 3.66 ^a,b^
***Cornus florida***	17.10 ± 6.89
***Cornus sanguinea***	130.03 ± 31.61 ^a,b^
**Ascorbic acid**	3.73 ± 0.83

^a^*p* < 0.05 vs. *Cornus florida*, ^b^
*p* < 0.001 vs. ascorbic acid.

**Table 5 plants-09-00122-t005:** IC_50_ values (µg/mL) of the extracts from fruits of *Cornus alba*, *Cornus florida*, and *Cornus sanguinea* for the inhibition of *α*-glucosidase and *α*-amylase.

	*Cornus alba*	*Cornus florida*	*Cornus sanguinea*
***α* -glucosidase**			
Aqueous-ethanolic extract	50.38 ± 14.05	38.87 ± 2.65	70.07 ± 16.62 ^†^
Acarbose	150.43 ± 0.04 *^,#^ 231.91 ± 0.06 (µM)		
***α*-amylase**			
Aqueous-ethanolic extract	115.20 ± 14.31 ^a,c^	5018.43 ± 14.70 ^b,c^	651.44 ± 12.99 ^a,b^
Acarbose	2.4 ± 0.06 ^§^ 3.7 ± 0.6 (µM)		

*α*-glucosidase: * *p* < 0.05 vs. *Cornus alba*; ^#^
*p* < 0.001 vs. *Cornus florida*; ^†^
*p* < 0.05 vs. *Cornus florida*; *α*-amylase: ^a^
*p* < 0.05 vs. *Cornus florida*, ^b^
*p* < 0.05 vs. *Cornus alba*, ^c^
*p* < 0.05 vs. *Cornus sanguinea*; ^§^
*p* < 0.001 vs. *Cornus alba*, *Cornus florida*, *Cornus sanguinea*.

**Table 6 plants-09-00122-t006:** The spectrometric data of compounds isolated from fruit of *Cornus alba* and IC_50_ values for inhibition.

No.	Fraction	UV Max	[M-H]^−^ *m*/*z*	MS^2^ Fragment Ions (*m*/*z*)	Identified Compound	IC_50_ [mM]	References
**1**	D2.1	217, 324	353	191	chlorogenic acid	0.80 ± 0.32	[18]
**2**	D2.3	223, 311	675 *	337, 191	5-*O*-coumaroylquinic acid	0.84 ± 0.14	[16]
**3**	D2.4	226, 312	351	163, 145, 119	5-*O*-(*E*)-*p*-coumaroylquinic acid methyl ester	0.16 ± 0.01	[17]
**4**	D2.5	265, 344	475	327, 299, 285, 255, 227	kaempferol 3-*O*-glucuronide 6″-methylester	0.21 ± 0.05	[19]
**5**	H1.1	196, 290	631 *	315, 179, 143	hydroxytyrosol glucoside	1.46 ± 0.18	[20]

* adducts.

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
