# Peer review of "Inhibition of Digestive Enzymes and Antioxidant Activity of Extracts from Fruits of Cornus alba, Cornus sanguinea subsp. hungarica and Cornus florida–A Comparative Study"

_plants, 2020, doi:10.3390/plants9010122_

Round 1

Reviewer 1 Report

Dear Editor,

The authors have answered all my concerns and hence can be accepted in its present form

Author Response

Dear Reviewer,

We would like to thank for the interest in our study.

Reviewer 2 Report

In this work the isolation of 9 (5) phytochemicals from Corni fructus is described. However the authors should also explain in more detail how did they choose which compounds to isolate (e.g. were they major peaks in the chromatograms, the active compounds in bioassays or something else).

The notation for statistical comparisons in in vitro assays should be uniform. The statistical comparison should also include standard inhibitors and it should be properly annotated in all the tables. E.g. in Table 4 it is not clear tom the annotation whether the activity of CA is different than CS. Furthermore, from the same table it seems that the activities of ascorbic acid and CF do not differ. It should be corrected in this and the other tables. If necessary the discussion should be modified so that it mentions only the statistically significant differences.

Supplementary material mentions only 4 isolated compounds while there are 5 compounds in Table 6 and the manuscript. Even though chlorogenic acid is a well known compound and it is not necessary to add its NMR spectrum, it should be added in the list after the isolation scheme.

Full names of the plant species should be added in the supplementary material

Author Response

Dear Editor and Reviewers,

Thank you for the suggestions required for the manuscript submitted to the Plants. We carefully reviewed the manuscript according to your suggestions. All changes were highlighted with “Track changes” function.

We would like to thank for the interest in our study and for comments that will help us to improve the manuscript. We have tried to do our best to respond to the points raised. According to the specific comments, we have revised the manuscript carefully and made necessary changes. Please find enclosed answers and rebuttals.

C1: In this work the isolation of 9 (5) phytochemicals from Corni fructus is described. However the authors should also explain in more detail how did they choose which compounds to isolate (e.g. were they major peaks in the chromatograms, the active compounds in bioassays or something else).

R1: Our aim was to isolate the most abundant compounds of C. alba fruits. Due to the technical aspects of laboratory practice we prepared the acetone-methanol-water extract for isolation procedures. We conducted the isolation with a standard available procedure of extraction using solvents with increasing polarity. The separation of compounds was tracked with TLC and HPLC methods at each step of isolation. Unfortunately, the separation of the most abundant compounds detected in the extracts was failed because of not relevant differences of retention times. Thus, only other minor compounds or artefacts were successfully isolated. We included some comments in this field in the manuscript (L. 171-176, L. 307-319).

C2: The notation for statistical comparisons in in vitro assays should be uniform. The statistical comparison should also include standard inhibitors and it should be properly annotated in all the tables. E.g. in Table 4 it is not clear tom the annotation whether the activity of CA is different than CS. Furthermore, from the same table it seems that the activities of ascorbic acid and CF do not differ. It should be corrected in this and the other tables. If necessary the discussion should be modified so that it mentions only the statistically significant differences.

R2: Thank you for this comment. We revised the statistical analysis and include the differences between extracts and positive controls. In fact, there is no statistical significance between DPPH scavenging activity of CF and ascorbic acid. The CA and CS activities were not statistically different. Therefore, we changed the footnotes in Table 4 and provided changes in the section 2.3 and Discussion (L. 182-190, L. 271- 285). The footnotes in Table 5 were also improved.

C3: Supplementary material mentions only 4 isolated compounds while there are 5 compounds in Table 6 and the manuscript. Even though chlorogenic acid is a well known compound and it is not necessary to add its NMR spectrum, it should be added in the list after the isolation scheme.

R3: Thank you for the comment. The chlorogenic acid was listed in the Supplementary Materials.

C4: Full names of the plant species should be added in the supplementary material

R4: It was changed.

Reviewer 3 Report

The article entitled “Inhibition of digestive enzymes and antioxidant activity of extracts from fruits of Cornus alba, Cornus sanguinea subsp. hungarica and Cornus florida – a comparative study” by Truba et al. well-designed manuscript with novel information.

However, the article requires some modifications and reviewer questions to address before consider to publish in plants

The authors used colorimetric DPPH radical scavenging assay to determine ani-oxidant capacity of tested samples. The assay results highly sensitive to color interferences. How authors avoid color interference of extract with final results?  

In advance it is better if authors can provide at least cyto-toxicity results of tested extracts suing cell culture assays.    

Minor corrections

Line 24: α-amylase, pancreatic lipase and α-glucosidase should revise as follows α-amylase, pancreatic lipase, and α-glucosidase (the general rule is A and B if more than two factors A, B, and C. Please recheck and revise accordingly)

Line 27: “DPPH scavenging assay, respectively..” should revise as follows DPPH scavenging assay, respectively.

Line 56: (e.g., flavonoids, anthocyanins)  => (e.g., flavonoids and anthocyanins)

Line 174: revise as follows diphenyl-picrylhydrazyl radical) scavenging ability (Table 4). The lowest half scavenging concentration (SC50) value, and

Line 407: were displayed as half scavenging concentration (SC50) values

In the table 4 and 5. Please provide full names of CA, CF, and CS.

In the table 6. Please provide full name of CA

The standard deviation reported in Line 188 and 189 shows 2 decimal units. However, IC50 values contains only one decimal unit. Please provide 150.43 ± 0.04 format.

Author Response

Dear Editor and Reviewers,

Thank you for the suggestions required for the manuscript submitted to the Plants. We carefully reviewed the manuscript according to your suggestions. All changes were highlighted with “Track changes” function.

We would like to thank for the interest in our study and for comments that will help us to improve the manuscript. We have tried to do our best to respond to the points raised. According to the specific comments, we have revised the manuscript carefully and made necessary changes. Please find enclosed answers and rebuttals.

The article entitled “Inhibition of digestive enzymes and antioxidant activity of extracts from fruits of Cornus alba, Cornus sanguinea subsp. hungarica and Cornus florida – a comparative study” by Truba et al. well-designed manuscript with novel information. However, the article requires some modifications and reviewer questions to address before consider to publish in plants.

C1: The authors used colorimetric DPPH radical scavenging assay to determine ani-oxidant capacity of tested samples. The assay results highly sensitive to color interferences. How authors avoid color interference of extract with final results?  

R1: In order to avoid color interferences, the blind samples were also tested. The absorbance of blind samples was included in the calculations. We have added a comment in L. 418. In general, the extracts did not interfere with absorbance signal of radical.

C2: In advance it is better if authors can provide at least cyto-toxicity results of tested extracts suing cell culture assays.  

R2: Thank you for the suggestion.   

Minor corrections

C1: Line 24: α-amylase, pancreatic lipase and α-glucosidase should revise as follows α-amylase, pancreatic lipase, and α-glucosidase (the general rule is A and B if more than two factors A, B, and C. Please recheck and revise accordingly).

R1: Thank you for this comment. It was improved.

C2: Line 27: “DPPH scavenging assay, respectively..” should revise as follows DPPH scavenging assay, respectively.

R2: Thank you for this comment. It was improved.

C3: Line 56: (e.g., flavonoids, anthocyanins)  => (e.g., flavonoids and anthocyanins)

R3: Thank you for this comment. It was improved.

C4: Line 174: revise as follows diphenyl-picrylhydrazyl radical) scavenging ability (Table 4). The lowest half scavenging concentration (SC50) value, and

R4: Thank you for this correction. It was improved.

C5: Line 407: were displayed as half scavenging concentration (SC50) valuesjj

R5: Thank you for this correction. It was improved.

C6: In the table 4 and 5. Please provide full names of CA, CF, and CS.

R6: It was changed.

C7: In the table 6. Please provide full name of CA

R7: It was changed.

C8: The standard deviation reported in Line 188 and 189 shows 2 decimal units. However, IC50values contains only one decimal unit. Please provide 150.43 ± 0.04 format.

R8: Thank you. It was changed.

Round 2

Reviewer 3 Report

Overall revised version of “Inhibition of digestive enzymes and antioxidant activity of extracts from fruits of Cornus alba, Cornus sanguinea subsp. hungarica and Cornus florida – a comparative study”, is addressed the reviewer questions and suggestions as expected.  Therefore, I’m suggesting to consider this manuscript to publish in Plants.